# RNA-Seq Dissects Incomplete Activation of Phytoalexin Biosynthesis by the Soybean Transcription Factors GmMYB29A2 and GmNAC42-1

**DOI:** 10.3390/plants12030545

**Published:** 2023-01-25

**Authors:** Jie Lin, Ivan Monsalvo, Melissa Ly, Md Asraful Jahan, Dasol Wi, Izabella Martirosyan, Nik Kovinich

**Affiliations:** 1Department of Biology, Faculty of Science, York University, Toronto, ON M3J 1P3, Canada; 2Department of Genetic Engineering and Biotechnology, Shahjalal University of Science and Technology, Sylhet 3114, Bangladesh

**Keywords:** glyceollin, transcription factor, *Glycine max*, *Phytophthora sojae*, wall glucan elicitor

## Abstract

Glyceollins, isoflavonoid-derived antimicrobial metabolites, are the major phytoalexins in soybean (*Glycine max*). They play essential roles in providing resistance to the soil-borne pathogen *Phytophthora sojae* and have unconventional anticancer and neuroprotective activities that render them desirable for pharmaceutical development. Our previous studies revealed that the transcription factors GmMYB29A2 and GmNAC42-1 have essential roles in activating glyceollin biosynthesis, yet each cannot activate the transcription of all biosynthesis genes in the absence of a pathogen elicitor treatment. Here, we report that co-overexpressing both transcription factors is also insufficient to activate glyceollin biosynthesis. To understand this insufficiency, we compared the transcriptome profiles of hairy roots overexpressing each transcription factor with glyceollin-synthesizing roots treated with wall glucan elicitor (WGE) from *P. sojae*. GmMYB29A2 upregulated most of the WGE-regulated genes that encode enzymatic steps spanning from primary metabolism to the last step of glyceollin biosynthesis. By contrast, GmNAC42-1 upregulated glyceollin biosynthesis genes only when overexpressed in the presence of WGE treatment. This is consistent with our recent discovery that, in the absence of WGE, GmNAC42-1 is bound by GmJAZ1 proteins that inhibit its transactivation activity. WGE, and not GmMYB29A2 or GmNAC42-1, upregulated the heat shock family gene *GmHSF6-1*, the homolog of Arabidopsis *HSFB2a* that directly activated the transcription of several glyceollin biosynthesis genes. Our results provide important insights into what biosynthesis genes will need to be upregulated to activate the entire glyceollin biosynthetic pathway.

## 1. Introduction

Phytoalexins are plant defense metabolites that are biosynthesized in response to pathogens [1,2,3]. Each plant species produces distinct phytoalexin metabolites that, collectively, represent molecules from all categories of specialized metabolism [4,5,6]. Their chemical diversity and roles as plant defense metabolites may be what affords them a wide range of biological activities in human cells [7]. The most well-known phytoalexins that are clinical drugs include the anticancer metabolite taxol [8], the antidiabetic berberine [9], and the antimalarial artemisinin [10]. Many other phytoalexins are in (pre)clinical testing [5,11,12], including the glyceollins from soybean that inhibit the survival and proliferation of several cancer cell and tumor types by mechanisms that are distinct from conventional therapeutics [13,14,15]. Additionally, glyceollins hold promise as natural antimicrobials in the swine industry [16,17], yet exhibit neuroprotective activities in mammalian cells [18]. All phytoalexins share one commonality: hey are produced at a low level in response to pathogens. Since phytoalexin biosynthesis is primarily regulated at the level of transcription of the biosynthesis genes [19,20,21], this feature makes the transcription factors that regulate those genes of prime importance as genetic targets for improving phytoalexin yields [21,22,23].

The regulation of phytoalexin biosynthesis is complex as it involves both positive and redundant, negative regulatory mechanisms, some of which were recently discovered to be conserved in regulating diverse phytoalexin biosynthetic pathways among different plant species (reviewed by [22]). In Arabidopsis, WRKY33 is a well-characterized transcription factor that directly positive regulates camalexin biosynthesis genes [20,24,25]. The phosphorylation of WRKY33 by the protein kinases MPK3/MPK6 and CPK5/CPK6 enhances its DNA binding and transactivation activities, respectively, and is needed for full activation of camalexin biosynthesis [26,27]. The ethylene response factor 1 (ERF1) interacts with WRKY33 and directly binds camalexin biosynthesis genes to mediate synergy between the ethylene, jasmonate, and map kinase (MAPK), signaling pathways in activating camalexin biosynthesis [28]. ERF72 is also phosphorylated by MPK3/MPK6 and physically interacts with WRKY33 to directly regulate camalexin biosynthesis genes [29]. WRKY33 directly represses ABA signaling genes, including *JAZ1*, and ABA suppresses camalexin biosynthesis [30]. RNAi silencing of *JAZ1* in Arabidopsis calli was recently shown to partially activate the biosynthesis of camalexin in the absence of an elicitor treatment [31]. Similarly, RNAi silencing *VvJAZ9* (the grapevine homolog of *JAZ1*) in grapevine cell cultures increases the production of the stilbene phytoalexin resveratrol [32]. By RNA-seq analyses of transcripts up and downregulated with glyceollin biosynthesis, we recently identified GmNAC42-1 (Glyma.02G284300) as the first transcription factor that is an essential and direct activator of glyceollin phytoalexin biosynthesis in soybean [33]. The Arabidopsis homolog of GmNAC42-1 was previously shown to be required for the activation of camalexin biosynthesis in *Arabidopsis* [34], yet it still remains unknown whether it directly regulates camalexin biosynthesis genes.

Transcription factors of the MYB family have not yet been found to directly regulate camalexin biosynthesis genes. However, the R2R3-MYB-type transcription factors VvMYB14 and VvMYB15 from grapevine (*Vitis vinifera*) directly regulate shikimate, phenylpropanoid, and stilbene synthase genes, the latter being specific to stilbene phytoalexin biosynthesis [35,36]. VvWRKY8 physically interacts with VvMYB14, inhibiting its interaction with VvSTS15/21 promoters to suppress stilbene biosynthesis [37]. By gene overexpression and silencing experiments in soybean hairy roots, we recently discovered that GmMYB29A2 (Glyma.02G005600), the soybean homolog of VvMYB14, is essential for activating glyceollin I phytoalexin biosynthesis in soybean and provides race-specific resistance to the pathogen *Phytophthora sojae* [2]. Like GmNAC42-1, GmMYB29A2 directly binds the promoter regions of glyceollin biosynthesis genes (i.e., *IFS2* and *G4DT*) in electrophoretic mobility shift and yeast one-hybrid assays [2,33]. Since these transcription factors directly activate the expressions of two genes that are required for glyceollin biosynthesis, they are prime candidates to engineer to fully activate glyceollin biosynthesis for stable production of those molecules. However, overexpressing either *GmNAC42-1* or *GmMYB29A2* without WGE treatment failed to activate the transcription of several glyceollin biosynthesis genes, resulting in no accumulation of glyceollins.

Glyceollin biosynthesis involves around 43 metabolic steps starting from glucose (Figure 1). In primary metabolism, D-glucose is transformed into D-erythrose 4-phosphate through the pentose phosphate pathway, followed by enzymatic condensation with pyruvate, oxidative cyclization, and dehydration to generate shikimate [38,39,40]. Shikimate is initially converted into L-phenylalanine, followed by reductive deamination and hydroxylation to generate 4-coumarate throughout the phenylpropanoid pathway [41,42]. In the isoflavonoid pathway, isoliquiritigenin is generated by condensation of 4-coumarate with CoA and 3-molonyl-CoA and transformed into daidzein. Daidzein is reduced and hydroxylated to produce (3R)-2′-hydroxydihydrodaidzein, the precursor of glycinol [33,43,44]. Finally, glyceollins I, II, and III are produced through the condensation of (-)-glycinol and prenyl diphosphate. The addition of prenyl diphosphate in the position 4 of (-)-glycinol is needed to produce glyceollin I, while the addition in position 2 for glyceollins II and III [45,46,47].

## 2. Results

### 2.1. Co-Overexpressing the Transcription Factors GmMYB29A2 and GmNAC42-1 Fails to Activate Glyceollin Biosynthesis in Absence of an Elicitor Treatment

Since *GmMYB29A2* and *GmNAC42-1* failed to activate glyceollin biosynthesis when each gene was overexpressed individually in the absence of an elicitor [2,33], we tested whether co-overexpressing both transcription factors together could activate glyceollin biosynthesis. *GmMYB29A2* and *GmNAC42-1* were cloned upstream of the *p35S* and *pFMV* viral promoters of the pGEMINI vector [48] and were co-overexpressed 26.6- and 45-fold, respectively, in mock-treated roots (Figure 2A). The co-overexpression resulted in a modest (1.2-fold) upregulation of *HIDH* and *G4DT* by qRT-PCR, whereas *C4H*, *IFS2*, and *I2′H* expressions were not upregulated. Glyceollin metabolite amounts also remained unchanged (Figure 2B). In contrast, co-overexpressing *GmMYB29A2* and *GmNAC42-1* 3.5- and five-fold in WGE-treated roots upregulated *I2′H* and *G4DT* 1.2- to 1.6-fold, respectively, and resulted in a 1.8-fold increase in the amounts of glyceollin metabolites (Figure 2A,B).

In summary, while co-overexpressing *GmMYB29A2* and *GmNAC42-1* increases glyceollin biosynthesis in WGE-elicited hairy roots, it is insufficient to drive glyceollin biosynthesis in the absence of an elicitor treatment. The observed results raised the question of the relative roles of WGE and these two transcription factors in reprogramming soybean genome expressions.

### 2.2. Comparative Transcriptome Analyses Identify Common and Distinct Gene Sets Regulated by WGE, GmMYB29A2, and GmNAC42-1

To gain insight into the relative roles of WGE, *GmMYB29A2* and *GmNAC42-1*, we compared the transcriptomes of WGE-treated empty-vector-transformed hairy roots that were biosynthesizing glyceollins to mock-treated hairy roots overexpressing *GmMYB29A2* or *GmNAC42-1* that were not biosynthesizing those metabolites. The WGE-treated hairy roots upregulated 3567 genes while downregulated 3567 genes (Figure 3A). Similarly, overexpressing *GmMYB29A2* 2.9-fold upregulated 3140 genes and downregulated 2926. However, overexpressing *GmNAC42-1* 115-fold only upregulated 890 and downregulated 403 genes. Of the genes that were upregulated by WGE, 765 and 534 were also upregulated by *p35S::GmMYB29A2* and *p35S::GmNAC42-1*, respectively, with 93 genes being upregulated in all three genotypes (Figure 3A, Appendix A). The latter gene-set included the transcription factor *GmMYB29A1* (Glyma.10G006600), that putatively has a role in promoting glyceollin turnover and/or competing biosynthetic pathways [2]. Of the WGE-downregulated genes, 571 and 102 were also downregulated in *p35S::GmMYB29A2* and *p35S::GmNAC42-1*, respectively, with 52 genes being downregulated in all three genotypes (Appendix A).

Ontological analysis of the genes up and downregulated by WGE found ‘Protein Binding’ and ‘DNA binding’ were the most highly represented functionalities in those gene-sets (Figure 3B). They were also most highly represented in *p35S::GmMYB29A2* and *p35S::GmNAC42-1* gene-sets (Figure 3C,D). Perhaps not surprisingly, a similar analysis of the genes commonly regulated by WGE and *p35S::GmMYB29A2*, WGE and *p35S::GmNAC42-1*, and all three genotypes demonstrated similar distributions of ontologies (Appendix A). This demonstrated that the WGE, *p35S::GmMYB29A2,* and *p35S::GmNAC42-1* broadly regulated genome functions in a proportionally similar manner. The ontology ‘Catalytic Activity’ comprised 8–12% of the downregulated genes in each genotype (Figure 3B–D). In contrast, ‘Catalytic Activity’ represented 7% of the genes that were upregulated by each genotype. These included glyceollin biosynthesis genes.

### 2.3. Overexpressing GmMYB29A2 or GmNAC42-1 in the Absence of an Elicitor Treatment Fails to Upregulate Several Glyceollin Biosynthesis Genes That Are Regulated by WGE

To understand how overexpressing *p35S::GmMYB29A2* or *p35S::GmNAC42-1* in the absence of an elicitor treatment fails to activate glyceollin biosynthesis, we conducted an in-depth comparison of biosynthetic gene expressions among mock-treated roots overexpression those transcription factors and empty vector hairy roots treated with WGE. We compiled a list of 338 genes annotated to encode enzymes spanning the 43 enzyme-catalyzed steps from *D*-glucose to the formation of glyceollins (Figure 1, Appendix A). This large number of genes was justifiable due to the palaeopolyploid genome of soybean [49], where several genes encode most enzymes. For example, there are 12 chalcone isomerase (CHI) genes and a similar number of chalcone synthase (CHS) genes with distinct expression profiles, subcellular localizations, and sub-functionalities [50,51]. WGE upregulated 87 genes for 33 of those enzymes (Figure 4A). The nine enzymes that WGE did not regulate were PGL, TKL1, DQS, and MEE32 from primary metabolism, EPSP, CM and ArAT from phenylpropanoid metabolism, and MTC and G2DT from the MEP and glyceollin pathways, respectively. 

Genes for four primary metabolism enzymes were upregulated by WGE but not by *p35S::GmMYB29A2* or *p35S::GmNAC42-1*. These were RPE, RPI, EMB1144, and ADT (Figure 4A). *p35S::GmMYB29A2* upregulated genes for most of the same primary enzymes as WGE, yet it also upregulated genes for PGL and CM. In contrast, *p35S::GmNAC42-1* upregulated only genes for HXK and SK, which WGE and *p35S::GmMYB29A2* also upregulated.

WGE upregulated genes encoding most phenylpropanoid and isoflavonoid pathway enzymes (Figure 4B). *p35S::GmMYB29A2* upregulated most of those enzymes except CHI. It also upregulated ArAT, which catalyzes an alternative route to L-phenylalanine that was not upregulated by WGE. In contrast, *p35S::GmNAC42-1* upregulated only genes for C4H and IFR, and downregulated genes for 4Cl, CHR, and HIDH. 

WGE upregulated genes for the vast majority of MEP and glyceollin I biosynthetic enzymes, including several monooxygenases that are candidates of glyceollin synthase (GLS) (Figure 4C). However, WGE did not upregulate the gene for G2DT, which is the branchpoint enzyme for glyceollin II biosynthesis [52]. *p35S::GmMYB29A2* upregulated G2DT and genes for all other WGE-regulated enzymes except 2′HR, CMK, and IDI2. The *p35S::GmNAC42-1*, in contrast, failed to upregulated most MEP and glyceollin genes. It upregulated only two G4DT-like genes and downregulated genes for PTS and GLS.

Since GmNAC42-1 has an essential role in activating glyceollin biosynthesis, we tested whether *p35S::GmNAC42-1* could upregulate glyceollin biosynthesis genes when roots were exposed to WGE, as previously reported [33]. Overexpressing *GmNAC42-1* 1.4-fold in WGE-treated hairy roots upregulates *IFS2* and *G4DT* 1.7- and 1.4-fold, respectively (Figure 4D). However, overexpressing *GmNAC42-1* 22.6-fold in mock-treated roots failed to upregulate both biosynthesis genes (Figure 4E), validating the RNA-seq results presented here (Figure 4C).

Thus, we observed that overexpressing *GmMYB29A2* in mock-treated roots upregulated genes for most, but not all enzymes that were upregulated by WGE treatment. In contrast, overexpressing GmNAC42-1 in mock-treated roots failed to upregulate the vast majority of those genes, even though it could upregulate some of them in elicited roots. Our results demonstrate that overexpressing *GmMYB29A2* and *GmNAC42-1* in the absence of an elicitor is insufficient to activate the transcription of all glyceollin biosynthesis genes, raising the possibility that one or more additional transcription factors are needed to activate the entire glyceollin biosynthesis pathway. 

### 2.4. The WGE-Upregulated Transcription Factor GmHSF6-1 Directly Activates the Expression of Biosynthesis Genes HIDH and I2′H

Of the 2364 genes that were upregulated exclusively by WGE, 11% (260 genes) were annotated as DNA binding transcription factors (Figure 3A). To search for additional activators of glyceollin biosynthesis, we compared those genes to our previously published RNA-seq datasets [2,33]. We narrowed down this list to 100 transcription factor genes by including only those transcription factors that were also significantly upregulated by WGE treatment in Harosoy63 seeds [2] (Appendix A). Finally, we reduced the list to 22 transcription factor genes by including only those also upregulated by acidity stress and downregulated by dehydration, which represent an elicitor and a suppressor of glyceollin biosynthesis, respectively [33]. Of those 22 genes, the heat shock family gene *GmHSF6-1* (Glyma.03G135800) was selected for functional characterization since it demonstrated homology to VIT_208s0007g08750 that is co-regulated with stilbene phytoalexin biosynthesis genes in grapevine [36].

The qRT-PCR confirmed that *GmHSF6-1* was upregulated by WGE treatment (Figure 5A). The *G. max* RNA-seq Atlas (https://soybase.org/, accessed on 20 December 2022) demonstrated that the expression of *GmHSF6-1* did not follow developmentally regulated patterns of *HIDH* or the isoflavonoid regulator *GmMYB176* and was more similar to *G4DT* and the glyceollin regulators *GmMYB29A2*, *GmNAC42-1* (Figure 5B). The predicted peptide sequence of GmHSF6-1 consisted of 231 amino acids and a three-component HSF DNA binding domain (DBD) (Figure 5C). Phylogenetic analysis demonstrated that GmHSF6-1 and four other soybean proteins clustered as a subgroup with the Arabidopsis proteins AtHSFB2a and AtHSFB2b (Figure 5D). The peptide had 55% and 49% similarity to the Arabidopsis proteins HSFA2a (AT5G62020) and HSFA2b (AT4G11660), respectively, which notably have roles in regulating biotic and environmental stress responses [53,54].

To assess the potential involvement of GmHSF6-1 in regulating glyceollin biosynthesis, the coding sequence was cloned from the cDNA of WGE-treated W82 hairy roots and was transferred downstream of the *p35S* viral promoter in the plant gene expression vector. Overexpressing *GmHSF6-1* 5.1-fold in mock-treated W82 hairy roots resulted in a 1.5-fold upregulation of GmNAC42-1, but this was not accompanied by any changes in the expression of five glyceollin biosynthesis genes and *GmMYB29A2* by qRT-PCR (Figure 5E). Metabolite measurements also found no difference in the total amounts of glyceollins, but the roots over accumulated their biosynthetic intermediate daidzein 1.7-fold compared to the empty vector control (Figure 5F). In contrast, overexpressing *GmHSF6-1* 4.8-fold in WGE-treated hairy roots resulted in a 1.7- to 3.1-fold upregulation of the biosynthesis genes *IFS2*, *HIDH*, and *I2′H* (Figure 5E) and a 2.2- to 3.6-fold increase in the amounts of glyceollin I, II, and III (Figure 5F). To investigate whether GmHSF6-1 is a direct regulator of glyceollin biosynthesis genes, we conducted yeast one-hybrid analysis by fusing GmHSF6-1 to the yeast Gal4 activation domain (AD) and by integrating the promoters of *HIDH* and *I2′H* upstream of the *HIS3* gene in the yeast genome. Both yeast strains expressing AD-GmHSF6-1 fusions could grow on medium lacking histidine (Figure 5G), indicating that GmHSF6-1 was physically binding those promoters. The results demonstrate that GmHSF6-1 is a direct activator of those glyceollin biosynthesis genes.

## 3. Discussion

The development of stable approaches for the production of phytoalexins has been ongoing for decades. These approaches include (semi)synthesis, culturing microbes that ectopically express phytoalexin biosynthesis genes in large bioreactors, combining elicitor treatments to elicit maximum biosynthesis in plant tissues, and overexpressing rate-limiting biosynthetic genes in plants [55,56,57]. Yet, each of these has suffered shortcomings [56,58,59]. In plants, the biosynthesis of phytoalexins is primarily regulated at the level of transcription of the biosynthetic genes [19,20,21]. This feature makes the transcription factors that regulate those genes of prime importance for enhancing phytoalexin biosynthesis [21,22,23]. However, only with the advent of next generation sequencing approaches, such as transcriptome sequencing (RNA-seq) and chromatin immunoprecipitation genome-wide sequencing (ChIP-seq), can we begin to understand how phytoalexin gene regulatory networks can be reprogrammed to unlock phytoalexin biosynthesis [22].

Our research has been focusing on understanding the control the glyceollin biosynthesis in soybean as a model phytoalexin gene regulatory network. We recently identified two essential activators of glyceollin biosynthesis, namely *GmMYB29A2* and *GmNAC42-1* [2,33]. RNAi silencing in soybean hairy roots demonstrated that two transcription factors were required to activate glyceollin biosynthesis and they were found to directly bind the promoters of essential biosynthesis genes, namely *G4DT* and *IFS2*, in vitro and in the yeast one-hybrid system. However, independently overexpressing each gene failed to activate glyceollin biosynthesis without an elicitor treatment [2,33]. In this study, we tested whether co-overexpressing both transcription factors could activate glyceollin biosynthesis in the absence of elicitation, but it could not (Figure 2). To understand why co-overexpressing both transcription factors could not activate glyceollin biosynthesis without elicitation, we compared the transcriptomes of WGE-treated empty-vector-transformed hairy roots that were biosynthesizing glyceollins to mock-treated hairy roots overexpressing *GmMYB29A2* or *GmNAC42-1* (Figure 2 and Figure 3). WGE upregulated 87 genes for 33 of the biosynthetic enzymes (Figure 4). This suggested that upregulating genes for these 33 enzymes may be needed to achieve glyceollin biosynthesis. Our results demonstrated that overexpressing GmMYB29A2 in mock-treated roots upregulated genes for all but eight enzymes that were regulated by WGE. These were for the primary metabolism enzymes RPE, RPI, EMB1144, and ADT, the isoflavonoid enzyme CHI, and the glyceollin/MEP pathway enzymes 2′HR, CMK, and IDI2 (Figure 4). Thus, overexpressing *GmMYB29A2* alone activates most, but not all genes needed to activate glyceollin biosynthesis. The current model comparing WGE versus GmMYB29A2 regulations is shown in Figure 6.

The transcription factor VvMYB14 from grapevine (*Vitis vinifera*) was recently reported to regulate biosynthetic steps for stilbene phytoalexin biosynthesis, including primary metabolism genes [36]. GmMYB29A2 shows the highest amino acid similarity to VvMYB14 among all grapevine proteins [2,35]. Additionally, genes encoding those two are syntenic with malate dehydrogenase on chromosomes 2 and 7 of soybean and grapevine, respectively. Together, this demonstrates that *GmMYB29A2* and *VvMYB14* are homologs. DNA affinity purification sequencing (DAP-Seq) combined with RNA-seq gene co-expression networks (GCN) analysis identified direct targets of VvMYB14 [36]. Those targets included biosynthesis genes that were also upregulated by *GmMYB29A2* overexpression, such as the primary metabolism genes for shikimate kinase (SK, biosynthetic step 13 in Figure 1) and chorismate mutase (CM, step 16), and phenylpropanoid pathway enzymes phenylalanine ammonia-lyase (PAL, step 21), cinnamate 4-hydroxylase (C4H, step 22) and 4-coumarate-CoA ligase (4CL, step 23) (Figure 4). However, our yeast one-hybrid and electrophoretic mobility shift assays (EMSAs) have shown that GmMYB29A2 directly binds the promoters of isoflavone synthase (*IFS*) and glycinol-4-dimethylallyltransferase (*G4DT*) [2], which have no obvious homologs in grapevine which does not biosynthesize glyceollins. Likewise, VvMYB14 directly binds stilbene synthase (STS) gene promoters in grapevine [35,36], and there have been no reports that soybean biosynthesizes stilbenoids. Thus, taken together with [36], our results strongly suggest that GmMYB29A2 and VvMYB14 have maintained roles in regulating several primary and early phenylpropanoid genes, but also have coopted distinct downstream genes for glyceollin and stilbene biosynthesis, respectively.

Since isoflavonoids and phenylpropanoids are the most abundant specialized metabolites that accumulate without stress in soybean and grapevine, GmMYB29A2 and VvMYB14 likely faced evolutionary pressure to coopt biosynthesis genes that could convert available metabolic intermediates into phytoalexins that are toxic to pathogens. The R2 R3 DNA binding domains of GmMYB29A2 and VvMYB14 are 89% similar/identical at the amino acid level (Appendix A), so the consensus DNA binding sequence bound by each factor is likely highly similar. DAP-seq suggested the VvMYB14-binding sequence [36]. However, VvMYB13, VvMYB14, and VvMYB15 have identical DNA binding domains and do not regulate identical gene-sets [36]. Thus, gene targets of VvMYB14-type proteins such as GmMYB29A2 are likely influenced by their interactions with other proteins, including other transcription factors. This could explain why GmMYB29A2 positively regulates *G4DT* for the biosynthesis of glyceollin I during elicitation [2], but instead regulates *G2DT* for the biosynthesis of glyceollin II in the absence of an elicitor (Figure 4).

Results on GmNAC42-1 suggest a similar scenario. GmNAC42-1 upregulated *IFS2* and *G4DT* when overexpressed in WGE-treated hairy roots (Figure 4D). However, it failed to upregulate those genes in the absence of WGE treatment (Figure 3E). The latter is consistent with our RNA-seq results, where overexpressing GmNAC42-1 without an elicitor failed to upregulate most glyceollin biosynthesis genes (Figure 3C). In a parallel study in our lab searching for genes that were oppositely regulated compared to glyceollin biosynthesis, we identify GmJAZ1-6 and GmJAZ1-9 as negative regulators of glyceollin biosynthesis [60]. These two proteins physically interact with GmNAC42-1 and block its activation of glyceollin biosynthesis gene promoters in the yeast three-hybrid and promoter-luciferase reporter systems [60]. Notably, silencing GmJAZ1s results in some accumulation of glyceollins in the absence of an elicitor. Thus, GmJAZ1 genes likely need to be silenced for GmNAC42-1 overexpression to activate most glyceollin biosynthesis gene targets. Despite that *p35S::GmNAC42-1* regulated numerous genes (Figure 2), only two primary metabolism genes that contribute to glyceollin biosynthesis were upregulated in the absence of an elicitor treatment (Figure 6). Thus, GmNAC42-1 likely has very different roles in the presence and absence proteins expressed in response to an elicitor treatment.

Unlocking the function of GmNAC42-1 by silencing GmJAZ1 gene expressions may be insufficient to activate all glyceollin biosynthesis genes, and additional transcriptional activators regulated by WGE may be required. The combinatorial action of GmMYB176 (Glyma.19g214900) and GmbZIP5 (Glyma.05G122400) controls the biosynthesis of developmentally regulated isoflavonoids in soybean [61] and GmMYB133 (Glyma.07G066100) modulates the biosynthetic pathway [62]. Yet, neither WGE, *p35S::GmMYB29A2*, or *p35S::*GmNAC42-1 upregulated those genes in W82 hairy roots (Appendix A). This study identified the WGE-upregulated GmHSF6-1 as a direct positive regulator of *HIDH* and *I2′H* (Figure 5). Thus, GmHSF6-1 likely cooperates with GmMYB29A2 to fully activate the expression of those genes.

GmHSF6-1 belongs to the heat shock factor family of transcription factors and compared to Arabidopsis proteins it shows highest similarity to HsfB2a (Figure 5). Plant HSFs of subgroup B are generally involved in non-heat shock functions [63]. HsfB2a positively regulates the expression of defensin genes in response to insects and microbial pathogens [64,65]. In vitro and agroinfiltration-based kinase assays have demonstrated that the Ca^2+^-dependent protein kinases (CPK3 and CPK13) phosphorylate HsfB2a and that is needed for full transcriptional activation of its defensin gene targets [64,65]. The same study demonstrated that CPK3 also phosphorylates ethylene response factor ERF1. In Arabidopsis upon pathogen infection, ERF1 functions to integrate ethylene and jasmonate (JA) signaling by directly activating camalexin phytoalexin biosynthesis genes [28]. ERF1 needs to interact physically with WRKY33 to directly bind and upregulate camalexin biosynthetic genes [28]. Homologs of both ERF1 and WRKY33 were found to be upregulated by WGE but not *p35S::GmMYB29A2* or *p35S::*GmNAC42-1 (Appendix A), indicating that they may be upstream of those transcription factors, if involved in activating glyceollin biosynthesis. We have discovered that GmNAC42-1- and GmJAZ1-type proteins have conserved roles in regulating glyceollin biosynthesis [33,60], as their homologs do in regulating camalexin biosynthesis in Arabidopsis [31,34]. Thus, HSF6-1 could be another member of a conserved transcription factor network that regulates distinct phytoalexin biosynthetic pathways among plant species.

## 4. Materials and Methods

### 4.1. Chemicals

Stocks (50 mg/mL) of the antibiotics including kanamycin, timentin, hygromycin-B, and ampicillin (Gold Biotechnology, Olivette, MO, USA) were prepared in MilliQ-purified water. Acetosyringone stock (Sigma-Aldrich, St. Louis, MO, USA) was 100 mM in dimethyl sulfoxide. (−)-Glyceollin I was purchased from Dr. Paul Erhardt (University of Toledo). Isoflavone standards were purchased from Extrasynthese (Genay, France). Other reactants were purchased from Sigma-Aldrich. Ultra-performance liquid chromatography (UPLC) solvents were liquid chromatography-mass spectrometry grade from Fisher (Hampton, NH, USA). WGE was extracted from race 1 *Phytophthora sojae* cultured on lima bean medium for three weeks as described previously [66].

### 4.2. Plant Materials

Soybean seeds of Williams 82 (W82) were obtained from the USDA-GRIN Soybean Germplasm Collection (Beltsville, MD, USA). The surface of W82 seeds (16–20 per batch) were sterilized with [70% isopropanol (*v*/*v*; 30 s), 6% sodium hypochlorite (*v*/*v*; 5 min), and distilled water (rinsed three times)] and imbibed overnight before use. For hairy roots, seeds were grown for 7 days [3 days at dark; 4 days under cool white T5 fluorescent lights (100 μEm^2^/s)] on germination and co-cultivation (GC) medium followed by the removal of the seed coat, and the distal end (~2 to 3 mm) of the cotyledons were excised as described by [2]. 

### 4.3. RNA Extraction and Gene Expression Measurements by qRT-PCR

Total RNA from soybean tissue was extracted using the Spectrum Plant Total RNA Kit (Sigma-Aldrich, St. Louis, MO, USA) following the manufacturer’s protocol. RNA-seq analysis was described in [2,33]. DNase I (Amplification grade, Invitrogen, Waltham, MA, USA) treatment of the total RNA (500 ng) was completed to remove genomic DNA contamination before complementary DNA (cDNA) synthesis, and SuperScript II Reverse Transcriptase (Invitrogen, Waltham, MA, USA) was used to synthesize cDNA following the manufacturers protocol. SsoAdvanced Universal SYBR Green Supermix (BioRad, Hercules, CA, USA) was used to carry out qRT-PCR of diluted cDNA templates as described previously [66]. Briefly, reactions (5 μL) consisted of 2 μL of first-strand cDNA (or untreated RNA), 0.5 μL of 5 mM forward and reverse primers, 0.5 μL of RNase free H_2_O, and 2 μL of the iQ SYBR Green Supermix (BioRad, Hercules, CA, USA). CFX Opus 96 Real-Time PCR System (BioRad, Hercules, CA, USA) was used to perform qRT-PCR on cDNA from three biological replicates or RNA that were not reverse transcribed to measure for genomic DNA contamination. The conditions of the qRT-PCR were as follows: initial denaturation at 95 °C for 3 min, followed by 39 cycles of 95 °C for 15 s and 60 °C for 30 s. Normalized gene expression was calculated based on cycle threshold (Ct) values using the formula expression = 2^ − [Ct(gene) − Ct(UBIQUITIN3)]. To verify the specificity of the qRT-qPCR reactions, melting curves were determined after each reaction. Primers used in this study are listed in Appendix A.

### 4.4. Transcriptome Data Analysis

The RNA sequencing and analysis of four biological replicates of WGE- and water-treated tissues of W82 hairy roots was reported in [2]. The data are available in the Gene Expression Omnibus (GEO) database at the National Center for Biotechnology Information (NCBI) under the accession number GSE131686. The same method from [2] was used to analyze three biological replicates of H_2_O-treated W hairy roots of p35S::GmMYB29A2, p35S::GmNAC42-1, and the p35S empty vector (pGWB2) control. Those data are available in GEO under the accession number GSE221901.

For analysis of glyceollin biosynthesis gene expressions, the enzyme and their corresponding Glyma gene accession numbers were obtained from the PlantCyc database of the Plant Metabolic Network (PMN), except glyceollin synthase (GLS). Cytochrome p450 monooxygenases [67] that were upregulated by WGE treatment were considered candidates of *GLS*. Any glyceollin biosynthesis gene that were significantly up or downregulated (adjusted *p* < 0.05) in any RNA-seq comparison (p35S::GmMYB29A2/vector, p35S::GmNAC42-1/vector, or WGE/mock), were compared by heat maps.

### 4.5. Cloning

The GmMYB29A2, GmNAC42-1, and GmHSF6-1 coding sequences were PCR amplified from the cDNA of WGE-treated Willliams 82 hairy roots and BP cloned into pDONR221. Primers are listed in Appendix A. For co-overexpression, the donor clones were simultaneously LR recombined into the dual overexpression vector pGEMINI [48]. GmHSF6-1-pDONR221 was LR cloned into pGWB2 and pDEST-GADT7 for overexpression and yeast one-hybrid assays. *HIDH* and *I2′H* promoter regions were cloned into the BsaI site of the pGG vector that is flanked by attL4 and attR1 recombination sites and LR recombined into the destination vector pMW2. 

### 4.6. Hairy Root Transformation and Elicitation

*Agrobacterium rhizogenes* strain K599 containing the empty vector and the overexpression constructs were used to transform soybean cotyledons using a previously described protocol [33] with slight modifications. Briefly, *A. rhizogenes* harboring the construct was grown overnight on plates containing Luria-Bertani (LB) medium, hygromycin, and kanamycin (50 mg/L), and cells from the plate were resuspended in phosphate buffer (pH 7.5) containing 100 μM acetosyringone to an OD_600_ of 0.5–0.8 before use. Soybean cotyledons were inoculated through several 1 mM-deep cuts on the adaxial surface of the cotyledon with a scalpel dipped in the *A. rhizogenes* suspension. The cotyledons were grown for 3 days under a 16 h photoperiod (~65 μE) in GC medium containing 100 μM acetosyringone and then transferred into hairy root growth (HRG) medium containing timentin (500 mg/L). Primary roots with secondary roots (2–3 cm) were grown for 5 to 7 days in HRG plates containing kanamycin and hygromycin (50 mg/L). Transgenic hairy roots (secondary roots that grew 3 to 6 cm on selection HRG plates) were harvested and cut into 1 cm pieces for WGE or water treatments. Secondary roots from a single primary root represented one biological replicate since each primary root is derived from a distinct transformation event. The excised 1 cm pieces of secondary roots were placed in HRG plates lacking antibiotics and then overlaid with 80 μL of WGE (20 mg/mL) or water. The hairy roots were then incubated for 24 h under cool white T5 fluorescent lights (500 μEm^2^/s). Treated roots were used directly for metabolite extractions or were harvested into liquid nitrogen and lyophilized for 3 to 5 d (BenchTop Pro, SP Scientific, Warminster, PA, USA) before storage at −80 °C for RNA extraction.

### 4.7. Metabolite Analyses

Treated roots (90–110 mg) were harvested and extracted according to [66]. The extracts were analyzed using a Vanquish UPLC system (Thermo Scientific, Waltham, MA, USA) configured with a Quaternary Pump F, Split Sampler FT Autosampler, and Diode Array Detector HL (DAD). The UPLC-DAD method and quantification of isoflavonoids using Chromeleon 7.2.10 software (Thermo Scientific, Waltham, MA, USA) was conducted according to [33]. Isoflavonoid amounts were measured from five biological replicates. Secondary roots from a single primary root represented one biological replicate since each primary root is derived from a distinct transformation event. Hairy roots for all treatments were generated from the same batch of seeds to avoid differences caused by the seed age. Roots harboring gene overexpression constructs and their corresponding empty vector controls were treated and harvested at the same time to prevent inter-day experimental variation.

### 4.8. Yeast One-Hybrid

Bait strains of *Saccharomyces cerevisiae* YM4271 (MATa, ura3–52, lys2–801, ade2–101, ade5, trp1–901, leu2–3, 112, tyr1–501, gal4D, gal80D, ade5::hisG) that integrate HIDH and I2′H promoter regions upstream of the *HIS3* gene were prepared as described by [33] and transformed with either pDEST-GADT7 or GmHSF6-1-pDEST-GADT7. Bait-pray strains were selected in media lacking both histidine and leucine and confirmed by PCR. Positive DNA-protein interactions were assessed by comparing differential growth in the presence of a competitive inhibitor of HIS3 enzyme (3-Amino-1,2,4-triazol, 3AT). Three biological replicates confirmed by two independent experiments are shown in results.

### 4.9. Statistics

To assess for any significant statistical differences between groups, the Tukey post hoc test in one-way ANOVA was applied between treatments and genotypes. Significant differences are shown as different letters in each box (at α = 0.05).

### 4.10. Assession Numbers

Accession numbers are as follows: GmMYB29A2, Glyma.02G005600; GmNAC42-1, Glyma.02G284300; GmHSF6-1, Gyma.03G135800; HIDH, Glyma.01G239600; I2′H, Glyma.15G156100. 

## 5. Conclusions

The elicitor molecule WGE from the pathogen *P. sojae* upregulates 87 putative enzyme-coding genes spanning primary metabolism to glyceollin biosynthesis in soybean hairy roots. Overexpressing the transcription factor GmMYB29A2 upregulated genes for most, but not eight enzymes. The transcription factor GmNAC42-1 fails to upregulate most genes in the absence of WGE, coinciding with its recently discovered physical interaction with the negative regulators GmJAZ1-6/9. We discovered that the WGE-upregulated transcription factor GmHSF6-1 is also a direct positive regulator of several glyceollin biosynthesis genes. Our current model (Figure 6) suggests that multiple activators and removing GmNAC42-1′s suppression by GmJAZ1-6/9 will be needed to activate the transcription of all glyceollin biosynthesis genes.

## Figures and Tables

**Figure 1 plants-12-00545-f001:**
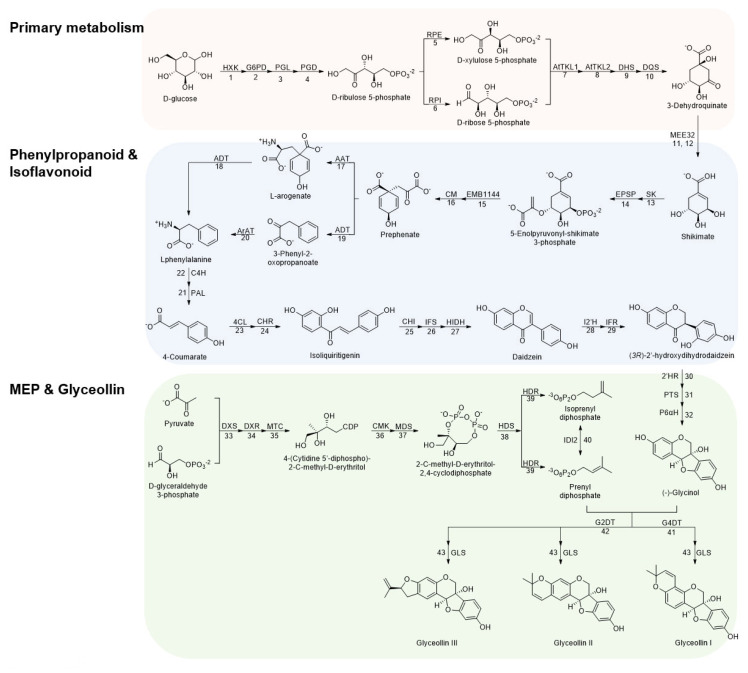
Glyceollin biosynthesis from D−glucose. All 43 enzymatic steps from primary metabolism, to phenylpropanoid and isoflavonoid pathways, to MEP and glyceollin biosynthesis are shown. HXK, hexokinase; G6PD, glucose−6−phosphate dehydrogenase; PGL, 6−phosphogluconolactonase; PGD, 6−phosphogluconate dehydrogenase; RPE, ribulose−phosphate 3−epimerase; RPI, ribose−5−phosphate isomerase; AtTKL1, Arabidopsis thaliana transketolase 1; AtTKL2, Arabidopsis thaliana transketolase 2; DHS, 3−deoxy−7−phosphoheptulonate synthase; DQS, 3−dehydroquinate synthase; MEE32, shikimate dehydrogenase; SK, shikimate kinase; EPSP, 3−phosphoshikimate 1−carboxyvinyltransferase; EMB1144, chorismate synthase; CM, chorismate mutase; AAT, L−glutamate:prephenate aminotransferase; ADT, arogenate dehydratase; ArAT, aromatic aminotransferase; PAL, phenylalanine ammonia−lyase; C4H, cinnamic acid 4−hydroxylase; 4CL, 4−coumarate−coenzyme A ligase; CHR, chalcone reductase; CHI, chalcone isomerase; IFS, isoflavone synthase; HIDH, 2−hydroxyisoflavanone dehydratase; I2′H, isoflavone 2′−hydroxylase; IFR, isoflavone reductase; 2′HR, (3R)−2′−hydroxyisoflavanone reductase; PTS, pterocarpan synthase; P6αH, dihydroxypterocarpan− 6α−hydroxylase; DXS, 1−deoxy−D−xylulose 5−phosphate synthase; DXR, 1−deoxy−D−xylulose 5−phosphate reductoisomerase; MTC, 2−C−methyl−D−erythritol 4−phosphate cytidylyltransferase; CMK, 4−(cytidine 5′−diphospho)−2−C−methyl−D−erythritol kinase; MDS, 2−C−methyl−D−erythritol 2,4−cyclodiphosphate synthase; HDS, (E)−4−hydroxy−3−methylbut−2−enyl−diphosphate synthase; HDR, 4−hydroxy−3−methylbut−2−en−1−yl diphosphate reductase; IDI2, isopentenyl−diphosphate Δ−isomerase; G4DT, trihydroxypterocarpan dimethylallyltransferase; G2DT, dimethylallylpyrophosphate:trihydroxypterocarpan dimethylallyl transferase; GLS, glyceollin synthase.

**Figure 2 plants-12-00545-f002:**
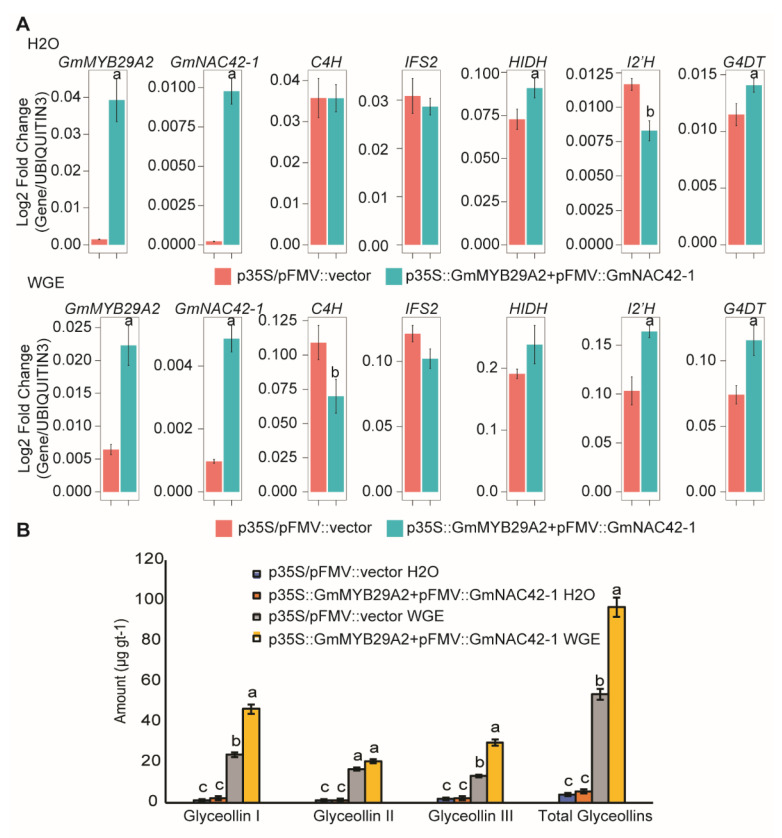
Co−overexpressing the transcription factors GmMYB29A2 and GmNAC42−1 fails to activate glyceollin biosynthesis in absence of an elicitor treatment. (**A**) Gene expressions in 24 h mock (H_2_O)−treated and *P. sojae* wall glucan elicitor (WGE)−treated hairy roots co−overexpressing *GmMYB29A2* and *GmNAC42−1*. The significance test was performed by paired students *t*−test, which is indicated by different letters (*p* < 0.05). (**B**) Amounts of glyceollin metabolites from hairy roots overexpressing *GmMYB29A2* and *GmNAC42−1* under mock or WGE treatment. The significance test was performed by single factor ANOVA, Tukey post hoc test, which is indicated by different letters (*p* < 0.01). Error bars represent SE (*n* ≥ 3).

**Figure 3 plants-12-00545-f003:**
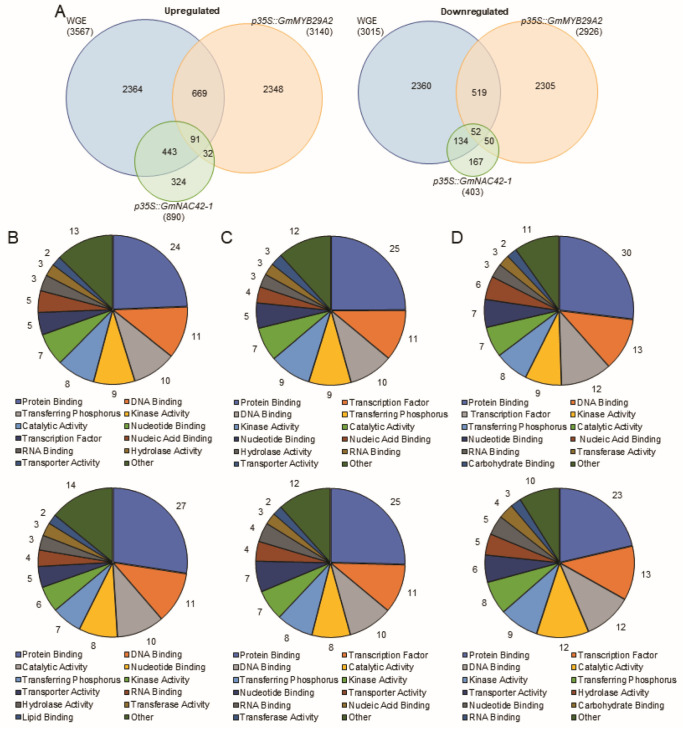
A comparison of differentially regulated genes among *P. sojae* wall glucan elicitor (WGE)-treated W82 hairy roots and mock-treated hairy roots overexpressing the transcription factors GmMYB29A2 and GmNAC42-1. (**A**) Venn diagrams of up and downregulated genes. (**B**) Percentage of ontologies of genes upregulated (top) and downregulated (bottom) by WGE. (**C**) Percentage of ontologies of genes upregulated (top) and downregulated (bottom) by p35S::GmMYB29A2. (**D**) Percentage of ontologies of genes upregulated (top) and downregulated (bottom) by p35S::GmNAC42-1. Ontology analysis was conducted using the (https://soybase.org/, accessed on 20 December 2022) Gene model Data Mining and Analysis tool.

**Figure 4 plants-12-00545-f004:**
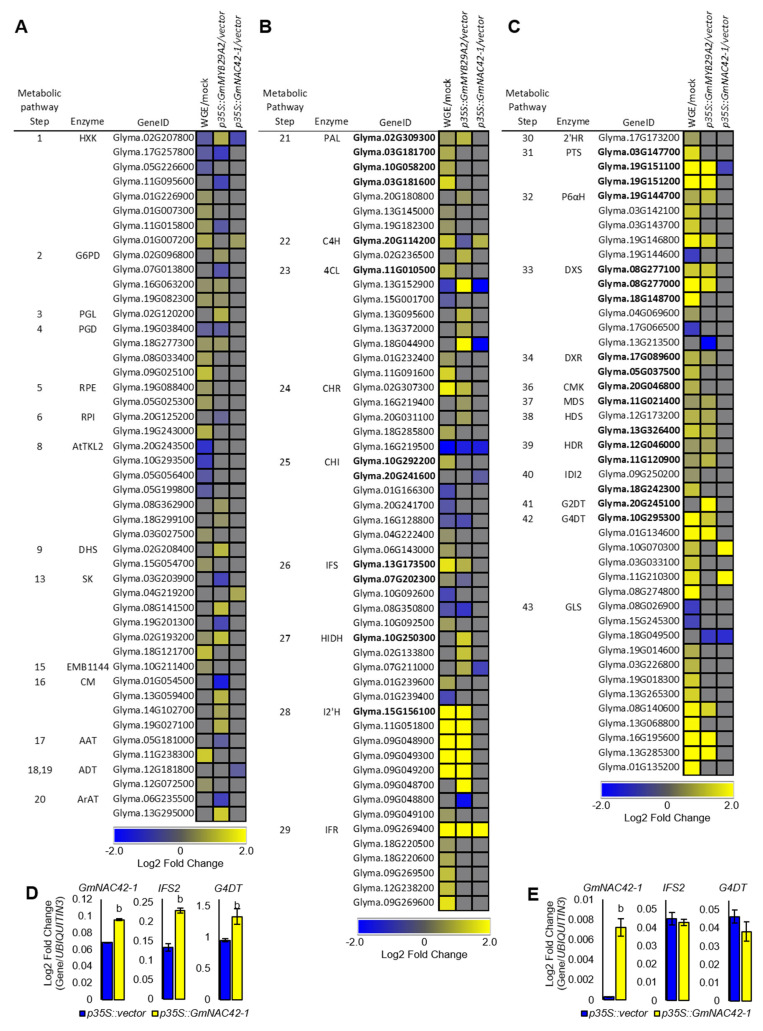
Glyceollin biosynthesis genes that were differentially regulated in soybean W82 hairy roots by *P. sojae* wall glucan elicitor (WGE), p35S::GmMYB29A2, and p35S::GmNAC42−1. (**A**) Heat map of primary metabolism genes corresponding to enzymatic steps 1–20 (see Figure 1). (**B**) Heat map of phenylpropanoid and isoflavonoid genes corresponding to enzymatic steps 21–29. (**C**) Heat map of the gene regulation from the methylerythritol phosphate (MEP) and glyceollin pathways (steps 30–43). (**D**) Gene expression levels in 24 h WGE−treated hairy roots of GmNAC42−1. (**E**) Gene expression levels in 24 h mock−treated (H2O) hairy roots GmNAC42−1.

**Figure 5 plants-12-00545-f005:**
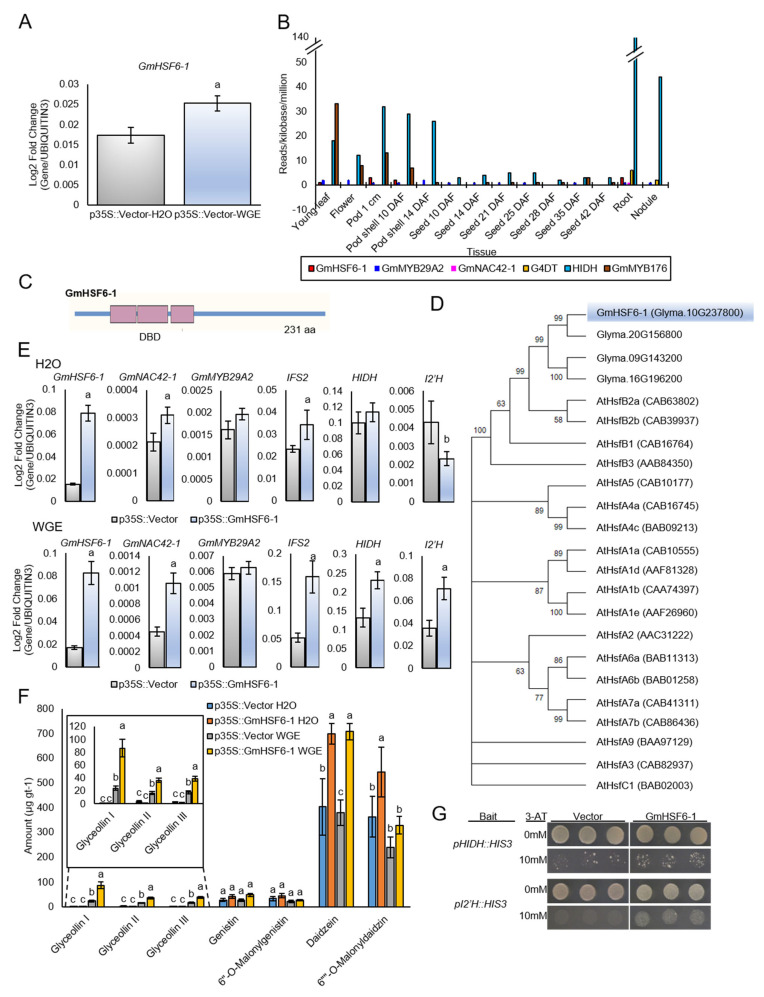
The *P. sojae* wall glucan elicitor (WGE)−upregulated transcription factor GmHSF6−1 directly activates the expression of glyceollin biosynthesis genes. (**A**) Expression level of *GmHSF6−1* gene in W82 hairy roots elicited for 24 h with WGE. (**B**) Gene expressions in developing soybean organs from the RNA−Seq Atlas of G. max (https://soybase.org/, accessed on 20 December 2022). GmMYB29A2, GmNAC42−1, and G4DT are markers of stress inducible glyceollin biosynthesis, whereas GmMYB176 and HIDH are markers of developmentally regulated isoflavonoids. (**C**) Schematic diagram of GmHSF6−1 gene indicating the DNA binding domain (DBD) region and coding sequence. (**D**) Cluster analysis of deduced amino sequences of GmHSFs with Arabidopsis AtHsfs. (**E**) Gene expressions in *p35S::GmHSF6−1* W82 hairy roots elicited for 24 h with H_2_O and WGE. ^a^ Significantly greater and ^b^ significantly less than control, paired students *t*−test (*p* < 0.01). Error bars represent SE (*n* ≥ 3). (**F**) Amounts of glyceollins and isoflavonoid metabolites from hairy roots overexpressing GmHSF6−1 24 h after treatment with WGE or H_2_O. Letters indicate significantly different levels determined by single factor ANOVA, Tukey post hoc test, *p* ≤ 0.01. (**G**) Yeast one−hybrid assays of YM4271 yeast transformed with GmHSF6−1 and HIDH/I2′H promoters on selection medium (SD−Leu−His+3AT). Representative values of two independent experiments are shown.

**Figure 6 plants-12-00545-f006:**
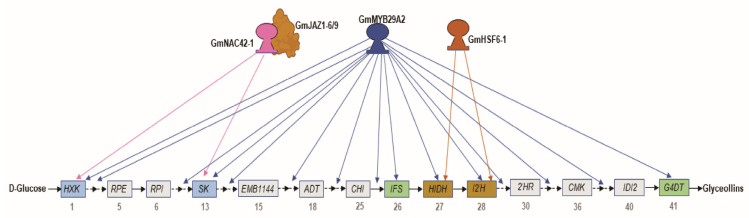
A schematic diagram of WGE-regulated glyceollin biosynthesis genes that are, and are not, upregulated by the transcription factors GmMYB29A2, GmNAC42-1, and GmHSF6-1 when each is overexpressed in the absence of an elicitor treatment. Solid and dotted black arrows indicate single and multiple enzyme-coding genes, respectively. Blue and pink arrows indicate genes that are upregulated by GmMYB29A2 and GmNAC42-1 via RNA-seq. Red arrows indicate genes that were upregulated by GmHSF6-1 by qRT-PCR. Genes that have been characterized to directly interact with GmMYB29A2, GmNAC42-1, and GmHSF6-1 are shown as blue, purple, and orange boxes, respectively. Note, only those genes have been assessed for protein DNA interactions. Gray boxes represent WGE-regulated genes that were not upregulated by overexpressing *GmMYB29A2*, *GmNAC42-1*, or *GmHSF6-1*. The interaction of GmJAZ1-6/-9 inhibits GmNAC42-1′s interaction with gene targets.

## Data Availability

All data supporting the finding of this study are available within the paper, Appendix A and the GEO under the accession number GSE221901.

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
