# Peer review of "RNA-Seq Dissects Incomplete Activation of Phytoalexin Biosynthesis by the Soybean Transcription Factors GmMYB29A2 and GmNAC42-1"

_plants, 2023, doi:10.3390/plants12030545_

Round 1
Reviewer 1 Report
This study proved that overexpression of transcription factors GmMYB29A2 and GmNAC42-1, either alone or combined, cannot upregulate the biosynthesis of glyceollins, a beneficial soybean metabolite. This finding overruled the hypothesis made based on previous research and has added new understandings to the biosynthetic pathway of glyceollins. I think the topic of the manuscript suits the journal and, based on my knowledge of the field (which might not be comprehensive), the research is novel and original. The writing is exceptionally well and straightforward. That’s why I suggested that it should be published as it was.
Author Response
Thank you for the encouraging comments, Reviewer 1!
Reviewer 2 Report
The manuscript entitled "RNA-seq dissects incomplete activation of phytoalexin biosynthesis by the soybean transcription factors GmMYB29A2 and GmNAC42-1" is well organized with high-quality presentation. It also provides valuable cues for the further dissection of the underlying mechanism fully activating phytoalexin biosynthesis. Due that the major results are based on transgenic hairy roots which is inheritable, I suggest the authors can go further with heritable transgenic soybeans. Here is also a minor confusion. In the introduction, "In Arabidopsis, WRKY33 is a well-characterized transcription factor that directly positive regulates camalexin biosynthesis genes" and "WRKY33 directly represses ABA signaling genes, including JAZ1, and suppresses camalexin biosynthesis", these two statements seem to conflict.
Author Response
We agree, making stable transgenic soybean should be the focus of future studies from our lab.
Regarding the conflicting statements, this was due to a typo in the second statement. We have corrected the second statement, adding the word ‘ABA’ as follows: ‘WRKY33 directly represses ABA signaling genes, including JAZ1, and ABA suppresses camalexin biosynthesis’
Thank you Reviewer 2 for carefully reading and catching this typo!
Reviewer 3 Report
The manuscript is well-designed and written, the objectives are clear, and the topic is worthy of investigation.
I have a few suggestions that might be useful and can improve the manuscript.
1. Explain the rationale to choose the heat shock family gene GmHSF6-1 for functional characterization.
2. The figure legend should be self-explanatory. Add full name for WGE in the legend of Figure 2-5.
3. The panel E in Figure 5 is too crowded for the y-axis label and figure legend.
Author Response
Reviewer 3 comment 1:
- Explain the rationale to choose the heat shock family gene GmHSF6-1 for functional characterization.
Response to Reviewer 3 comment 1:
We added the following, ‘Of those 22 genes, the heat shock family gene GmHSF6-1 (Glyma.03G135800) was se-lected for functional characterization since it demonstrated homology to VIT_208s0007g08750 that is co-regulated with stilbene phytoalexin biosynthesis genes in grapevine [36].’
Reviewer 3 comment 2:
- The figure legend should be self-explanatory. Add full name for WGE in the legend of Figure 2-5.
Response to Reviewer 3 comment 2:
Corrected.
Reviewer 3 comment 3:
- The panel E in Figure 5 is too crowded for the y-axis label and figure legend.
Response to Reviewer 3 comment 3:
Corrected.
Thank you for the helpful comments Reviewer 3. They have truly helped improve the clarity and quality of our manuscript.